# Narrowband Thermal Terahertz Emission from Homoepitaxial GaAs Structures Coupled with Ti/Au Metasurface

**DOI:** 10.3390/s23104600

**Published:** 2023-05-09

**Authors:** Ignas Grigelionis, Vladislovas Čižas, Mindaugas Karaliūnas, Vytautas Jakštas, Kȩstutis Ikamas, Andrzej Urbanowicz, Marius Treideris, Andrius Bičiūnas, Domas Jokubauskis, Renata Butkutė, Linas Minkevičius

**Affiliations:** 1Center for Physical Sciences and Technology, Saulėtekio Ave. 3, 10257 Vilnius, Lithuania; 2Institute of Applied Electrodynamics and Telecommunications, Vilnius University, Saulėtekio Ave. 3, 10257 Vilnius, Lithuania; 3Institute of Photonics and Nanotechnology, Vilnius University, Saulėtekio Ave. 3, 10257 Vilnius, Lithuania

**Keywords:** metasurface, thermal emission, magnetic polaritons

## Abstract

We report on the experimental evidence of thermal terahertz (THz) emission tailored by magnetic polariton (MP) excitations in entirely GaAs-based structures equipped with metasurfaces. The *n*-GaAs/GaAs/TiAu structure was optimized using finite-difference time-domain (FDTD) simulations for the resonant MP excitations in the frequency range below 2 THz. Molecular beam epitaxy was used to grow the GaAs layer on the *n*-GaAs substrate, and a metasurface, comprising periodic TiAu squares, was formed on the top surface using UV laser lithography. The structures exhibited resonant reflectivity dips at room temperature and emissivity peaks at T=390 °C in the range from 0.7 THz to 1.3 THz, depending on the size of the square metacells. In addition, the excitations of the third harmonic were observed. The bandwidth was measured as narrow as 0.19 THz of the resonant emission line at 0.71 THz for a 42 μm metacell side length. An equivalent LC circuit model was used to describe the spectral positions of MP resonances analytically. Good agreement was achieved among the results of simulations, room temperature reflection measurements, thermal emission experiments, and equivalent LC circuit model calculations. Thermal emitters are mostly produced using a metal-insulator-metal (MIM) stack, whereas our proposed employment of *n*-GaAs substrate instead of metal film allows us to integrate the emitter with other GaAs optoelectronic devices. The MP resonance quality factors obtained at elevated temperatures (Q≈3.3to5.2) are very similar to those of MIM structures as well as to 2D plasmon resonance quality at cryogenic temperatures.

## 1. Introduction

The terahertz (THz) frequency range of the electromagnetic wave spectrum is of particular research interest due to its unique properties [1,2,3]. The spectral fingerprints of resonant vibrations of molecules fall in the THz frequency range. It makes THz technology a perfect tool for imaging and spectroscopy [4], especially for spectroscopic imaging of scattering objects. It can penetrate many non-conductive materials, such as paper, wood, and plastic, making it useful for imaging concealed objects that are difficult to study with other forms of radiation. THz spectroscopy can be used to identify the unique spectral features of different materials, which can be useful for imaging and identifying hidden objects [5] and for medical diagnosis [6].

In general, the cost of a THz imaging system will depend on the type of setup and specific components. For example, pre-made components such as commercially available 200 nm gate-length GaAs high-electron-mobility transistors (HEMTs) as a cheap THz solution for 94 GHz detection were presented recently [7]. Moreover, diffraction gives the opportunity to manipulate a phase shift of radiation that passes through specific designed optical elements [8] or a variety of diffractive optics [9]. The precise control over the phase shift allows for almost free transform of an incident wavefront [10] or beamforming performance [11]. Planar structures may also be obtained using metamaterial designs [12]. A novel state-of-the-art THz source configuration of system-on-a-chip for computational THz imaging that combines a high power incoherent THz light generation with fast amplitude and frequency encoded spatial modulation was demonstrated [13]. There also exists a wide technological variety of THz emitters, including ones based on direct generation from a DC power supply (e.g., quantum cascade lasers (QCLs), difference frequency generation QCLs (DFG-QCLs), resonant-tunneling diodes (RTDs), HEMT, Gunn diodes, complementary metal oxide semiconductor (CMOS) voltage-controlled oscillators, etc.) [14,15,16] or with the generation based on manipulation of the external microwave or light source (e.g., frequency multiplication, surface current excitation, or spintronics employment by ultra-short pulses, etc.) [17,18,19]. Being of very different types, the majority of these emitters, however, have a common disadvantage: the emitters that are able to generate THz radiation in room temperature conditions usually either have a complicated inner structure or require additional highly priced and bulky components. This results in the lack of cheap, compact frequency tunable sources and in the high demand of extension of the THz emitters family. The concept of a thermo-based narrow band THz emitter is of high interest. It relies on heating a material to a high temperature and then using an antenna or other type of element to tailor the broadband thermal radiation in the THz frequency range and to couple it out from the device effectively. These elements can have a relatively simple inner structure and be inexpensive to build and convenient to use.

One of the viable ways would be to employ surface polaritonic excitations that, unlike, for example, widely used resonant 2D plasmon excitations in THz frequencies [20,21,22], are inherently free of limitations set by cryogenic operating temperatures and the submicron features of grating couplers [23], therefore not requiring advanced fabrication techniques and being more convenient to use. It is common to produce polaritonic resonators using metal-insulator-metal (MIM) configurations [24,25] or to employ alternative approaches such as doped GaAs quantum wells embedded in MIM structures instead of insulator layers [26]. However, no such devices compatible with the III-V semiconductor technology widely used in THz optoelectronics [9,27,28] has been demonstrated so far.

In this work, we present a thermal THz emitter based on magnetic polariton (MP) excitations in a *n*-GaAs/GaAs/metal stack providing narrow emission peaks in the range below 2 THz. In this stack, the cavity is formed in the insulating GaAs layer between the *n*-GaAs substrate and a TiAu metasurface. In such a cavity, polaritonic modes can be excited, which tailor broadband thermal emission spectra in THz region. This type of emitter has a number of potential advantages, including high-temperature operation and simple fabrication. The use of a metal metasurface also allows for precise control of the THz radiation frequency, as the geometry of the metasurface can be easily modified while device temperature control can be used to vary the radiation intensity. Such a low-cost, frequency and intensity variable solution is of special interest for the wide variety of applications, including, for example, THz imaging or short range THz data links.

## 2. Materials and Methods

The geometrical parameters of the metasurface used in this work are optimized according to the finite-difference time-domain (FDTD) simulations to set the resonant frequency position and the optimal conditions for efficient MP excitation. For the proof-of-concept, square metacells were selected in order to achieve the resonance of the same frequency across both directions of the metacell [24]. Having the device structure defined as *n*-GaAs substrate/GaAs spacer/TiAu metasurface, there are five variable structure parameters: spacer (*d*) and metasurface layer height (*h*), spacer doping, metacell period (*L*), and metacell side length (*l*). The output to be analysed was resonant frequency and power of the emitted radiation. Figure 1 represents the simulation results depicting resonant frequency and power at resonant frequency dependency on the period of the metasurface (*L*) and square metacell side width (*l*) (spacer height d=4.3 μm). One may note that the resonant frequency grows with an increase in the metasurface side length. Frequencies as low as 0.3 THz are available; however, the power is expected to decrease significantly. Reducing the side length results in emission at frequencies of up to 3 THz. Furthermore, it is clearly visible that the resonant frequency does not change with the period, meaning that, as will be explicitly described later, these simulations expose the effect that has the nature of magnetic polaritons. On the other hand, the period and spacer height also (not shown in the pictures) have effect on the power at the resonant frequency. Thus, in order to achieve a well-optimized thermal metamaterial emitter on the desired resonant frequency, all its structural parameters should be taken into account.

As the result, four samples (parameters marked by white circles on Figure 1) with different lateral dimensions of square metacells were produced, and the resonant response in room temperature reflectivity and emission at T = 390 °C in the range from 0.7 THz to 1.3 THz was demonstrated. In addition, the experimentally obtained resonance quality factor Q≈3.3 to 5.2 is similar to the values obtained for thermal emitters with MIM structure [24] as well as for 2D plasmons in heterostructures at cryogenic temperatures [21,29].

Along well-known polariton excitation schemes of attenuated total reflection or surface grating, it was demonstrated that such phenomena can arise in a bilayer of single-negative materials [30] composed of the metallic metasurface/insulator layer (μ<0,ε>0) and conductive plane layer (μ>0,ε<0) [31,32]. Polaritons excited in such a system are referred to as magnetic polaritons [32,33,34,35], as the system demonstrates diamagnetic behaviour. A single metasurface patch and the area of the conductive plane under it can be approximated as a pair of conductive patches [32]. The varying magnetic field component of the electromagnetic wave would induce the current loop in a patch pair, which, according to Lenz’s law, induces a magnetic field opposing the extrinsic one. The coupling of electromagnetic wave with MP would lead to an increased absorption at the resonant polariton frequency ($\omega_{r}$). The MPs are localized between conducting patches, and $\omega_{r}$ depends on the thickness of the insulation layer and the geometrical width of the patch. In general, the number of current loops between patches corresponds to the number of anti-nodes of magnetic field induction, as illustrated by modelling results depicted in insets (b) and (c) of Figure 2 for the main and the third harmonic of MPs, respectively. The current direction in the loops in two adjacent anti-nodes is anti-parallel (denoted by arrows in Figure 2c); therefore, their net magnetic moment *m* is zero. This implies that, if there is an even number of anti-nodes in the patch length, the total magnetic moment *m* is zero, and no magnetic resonance can occur. In the case of an odd number of anti-nodes, one of them stays unpaired, m≠0, and magnetic resonance occurs. However, this is true only for symmetrical conditions, when the angle of incidence (θ) equals zero. If there is an inclined incidence, even modes of MP can also exist.

The *n*-GaAs/GaAs/metal structure is shown in Figure 2. The undoped 4.3-μm-thick GaAs layer was grown on the 525-μm-thick *n*-type GaAs substrate using the molecular beam epitaxy technique. The electron density *N* measures $2\;\times \;10^{18}$ cm${}^{-3}$ in the substrate, resulting in a plasma frequency $\omega_{p}=\sqrt{Ne_{c}^{2}/\left(\varepsilon_{0}\varepsilon_{d}m^{*}\right)}\approx 2\pi \times 13$ THz, where $\varepsilon_{0}$ stands for dielectric permittivity in vacuum and $\varepsilon_{d}$ is the permittivity of undoped GaAs. The obtained plasma frequency value sets the upper limit of the device operational frequency. Indeed, $\omega_{p}$ can be varied with the doping density *N*, giving the additional degree of freedom when setting the emitter frequency bandwidth. The TiAu metasurfaces were fabricated using maskless UV laser lithography and lift-off process of sputtered thin metal film. Direct laser write system DWL66+ (Heidelberg Instruments Mikrotechnik GmbH, Heidelberg, Germany) and image-reversal AZ5214 (Merck Performance Materials GmbH, Wiesbaden, Germany) photoresist were used to form a pattern for metal film evaporation. Metallization layer comprising of 20 nm of Ti and 180 nm of Au was evaporated using an TFDS-870 (VST Service Ltd., Petah Tikva, Israel) e-beam sputtering system, and the structures were formed by a lift-off process of this film. The image of processed metasurface is shown in inset (a) of Figure 2, where one can notice the rounding of the corners of square metacells. The influence of corners rounding to the MP resonance frequency was simulated using a finite-difference time-domain method. The results show that the scale of rounding evident after sample processing had an intangible effect, reaching no more than 2% for the blueshift in resonant features of the devices and less than 1% for amplitude variation.

In order to support the last claim and to further disclose the simplicity of the proposed thermal metamaterial emitter, additional simulation was performed to evaluate the importance of the squareness of the metacell. In order to achieve that, right angles of the 42μm width metacell were changed to circles with increasing radius *R*, resulting in square metacell (R=0μm) conversion into circular metacell (R=l/2) (cf. bottom right inset of Figure 3). As one may see in the main part of Figure 3, there is a small resonant frequency increase within this conversion; however, it is still worth noticing that for the case of fabrication inaccuracies (R≪l/2), the resonant frequency change is very small. Furthermore, spectral dependencies on the upper left inset of Figure 3 depict a small increase in absorption with the increase of radius (black arrow depict the radius increase, bold black spectrum corresponds to the black point on the main graph, corresponding to the radius of (R=5μm), depicted at the center of the bottom right inset). The results show that the proper thermal emitter may be achieved employing considerably low quality fabrication equipment, which allows for a decrease in the final product price and an increase in its applicational attractiveness.

## 3. Results

In Figure 4, the reflectance and emissivity spectra are shown for different metasurface cell side lengths. The reflectance signal was recorded at room temperature using a THz time domain (THz-TDS) setup modified for the measurements at the angle normal to the sample surface, as depicted in the inset of Figure 4a. Here, the 50:50 beamsplitter (a high resistivity 525 μm-thick silicon plate) was placed at the optical path between the sample and the photoconductive antenna THz source, which directed the portion of the beam reflected from the sample surface to the photoconductive antenna THz detector. The flat gold mirror was placed instead of the sample for reference. One can see the reflection resonance at the lower frequency end, which blueshifts from 0.74 THz to 1.32 THz with a change of metacell side length from 42 μm to 26 μm, respectively, while the period of metacells is kept L=50 μm in all cases. The full width half maximum (FWHM) also slightly changes from 190 GHz to 255 GHz, giving resonance quality factor *Q* values from 5.2 to 3.3, respectively. In the frequency range from 2.1 THz to 2.5 THz, the third harmonic of MPs are visible for larger metacells, whereas for the 26-μm-size metacell, the third harmonic peak lies beyond the 3 THz limit of the THz-TDS setup. The traces of water vapour absorption (dips at 1.65 THz and 2.20 THz) are also present in the spectra because the experiment was performed in an ambient atmosphere. The reflectivity dip at 2.20 THz overlaps with the third harmonic MP resonance line; therefore, it appears broader than it actually is.

In Figure 4b, the emissivity spectra of the samples heated up to 390 °C are shown. The emission was measured with a custom-made Fourier transform spectrometer with a Golay cell as a detector (see inset of Figure 4b). The samples were placed on an external heater, ensuring the proper heat transfer to the sample, and the heating was managed using an electronic controller. To avoid background radiation, the heater and samples were placed in a non-radiating box with a 4 mm conical output aperture. A low-pass 4 THz filter diminished the unnecessary higher-frequency thermal radiation. The measurements were performed at a right angle to the metasurface. The reference thermal emission of the *n*-GaAs/GaAs structure without top metallization was also recorded in the same conditions. As can be deduced from Figure 4b, the emissivity results in the resonant spectral line with the peak positions shifted slightly to the low-frequency side compared to reflectivity spectra (Figure 4a). The main harmonic emission peaks shift from 0.71 THz to 1.22 THz for the metacell sizes from 42 μm to 26 μm, respectively. The FWHM of the emission bands broadens from 280 GHz to 350 GHz. Similar to reflectance results, the peaks corresponding to the third harmonic of the MPs are also visible at the higher energy side of the spectra, and no plasmon polariton related features are manifested. Moreover, spectra are modified by water vapor absorption.

It is also important to add that the observed peaks are not related to surface plasmon polaritons, as in this case the position of the resonance would change with the period *L* rather than with the square side *l*. Moreover, the magnetic polaritons emission, unlike the emission of surface plasmon polaritons, does not have angular dependencies. In order to verify this, the angular emission of the main harmonic from the sample equipped with 26 μm × 26 μm squares containing the metasurface was measured at a few different incidence angles θ (see the inset in Figure 5). The experimental results of the main harmonic peak at incidence angles of 0,8, and 23 degrees are presented in Figure 5. As can be seen from Figure 5, resonant peaks for 0 deg and 23 deg coincide well with each other at f=1.22 THz, whereas the central frequency for θ=8 deg is at ≈1.20 THz. Therefore, it can be deduced that resonant peak position has no dependency on the rotation angle, and it represents magnetic polariton excitation. It is clearly seen that the width of the spectral features is identical for θ=0 deg and θ=23 deg, whereas it increases significantly for θ=8 deg, which might be a result of non-optimal sample alignment along the optical axis. The solid curve represents the 2D rigorous coupled wave analysis (RCWA) [36] simulated spectrum for zero angle of incidence. It is shifted to lower frequencies with respect to experimental data (see Table 1); however, its width is very close to the experimentally obtained one. The RCWA simulations for non-zero angle of incidence used in the experiment gave identical results as in case of θ=0; therefore, the corresponding curves are not shown.

The features representing the second harmonic of the resonance do not appear in the experimental spectra. This may be related to MP formation peculiarities at incidence angle θ=0 deg, as discussed earlier. The appearance of only odd harmonics is also confirmed by 2D RCWA simulations presented by color in Figure 6. In the simulations, the above-described structure geometrical parameters were taken, and the dielectric function included possible plasmonic and phononic influence. In addition, the non-polarized electromagnetic radiation, incident normal to the sample surface, was considered. The frequency dependencies of n=1,3,and5 order MP branches on the metacell square side length are depicted as darker color (orange) traces. In addition, the experimental positions of the room temperature reflection and T=390 °C emission resonances are marked as diamonds and circles, respectively. Good agreement between the simulations and the reflectivity experiment is demonstrated. However, the positions of resonances observed in emissivity are tuned to lower frequencies by less than 8 percent, most probably because of GaAs dielectric function modification due to elevated temperature [37].

## 4. Discussion

In order to confirm the excitation of magnetic polaritons in our proposed emitter structure, we have used the equivalent 1D LC circuit approach, which allows an analytical estimation of MP resonance frequencies in metasurface/dielectric/metal film structures [32]. The equivalent circuit is depicted in the inset of Figure 6. The impedance *Z* of such a circuit is calculated as:(1)$$Z=\frac{j\omega (L_{m}+L_{e})}{1-\omega^{2}C_{e}\left(L_{m}+L_{e}\right)}-\frac{2j}{\omega C_{m}}+j\omega \left(L_{m}+L_{e}\right),$$
where $L_{m}=0.5\mu_{0}d$ is Faraday inductance and $L_{e}\;=\;1/\gamma h\omega_{\mathrm{Aup}}^{2}\varepsilon_{0}$ is kinetic electron inductance in the pair of conductive patches; $\mu_{0}$ is magnetic permeability, γ is the numerical factor describing the effective cross-sectional area of the metal square patch in a vacuum, and $\omega_{\mathrm{Aup}}\;=\;1.32\times 10^{16}$ Hz is the plasma frequency in gold; $C_{m}\;=\;A\varepsilon_{d}\varepsilon_{0}w^{2}/d$ is the capacitance between conductive patches, and $C_{e}\;=\;\pi \varepsilon_{0}w/\log\left(L-w/h\right)$ is the capacitance of the air gap between two adjacent metacells. Here, *A* is the factor accounting for the effective area of the capacitor in the metal patch. The resonant frequency ω of the LC circuit is then given by:(2)$$\omega_{r}={\left(\frac{C_{m}+C_{e}-\sqrt{C_{m}^{2}+C_{e}^{2}}}{\left(L_{e}+L_{m}\right)C_{e}C_{m}}\right)}^{\frac{1}{2}}.$$

The calculated MP resonance frequency dependencies on the square metacell side length for the first and third harmonics are shown by the solid curves in Figure 6. Choosing the numerical factor values of A=0.35 and γ=0.66 gave the best agreement with experimental and RCWA-simulated data. The effective patch cross-sectional area (γ=2/3) is the same as found elsewhere [32]; however, *A* is slightly different from reported 0.2⩽A⩽0.3 in the literature [38]. We assume this is because, in our case, a 2D metasurface is considered, where the effective area of the capacitor can be influenced by a higher number of neighbouring patches than in the 1D case. In addition, there is a significant mismatch between calculated LC resonance and RCWA simulations for l<30 μm. Here, numerical simulation data show that the third and fifth MP harmonics tend towards each other what should cause interaction between these mode, and such effect is not accounted for in the LC equivalent circuit model. Nevertheless, good agreement is achieved between existing experimental results, RCWA simulations, and LC equivalent circuit calculations. In Table 1, the resonant frequencies estimated using different methods are listed for comparison.

In order to compare the performance of the proposed emitter with the other compact and solid state based THz emitters, the emission optical power of the first harmonic MP peak of the sample equipped with l=26 μm squares metasurface was calculated. At the most conservative estimation it reaches ∼200 nW when the temperature of the sample is 390 °C. In order to reach such temperature, the external heater required electrical power of 40 W; therefore, the proposed emitter’s wall plug efficiency (WPE) is approximately $5\times 10^{-9}$. The reported thermal emission power at f=5 THz from ultrastrongly coupled polaritons in metal-insulator-metal structure with multiple AlInAs/GaInAs quantum wells was 4.8 nW at 300 K and 2.3 nW at 4 K [26]. In this case, the structure was fed with average electrical power of 1.6 W, thus leading to a WPE of $3\times 10^{-9}$ and $1.5\times 10^{-9}$ at T=300 K and 4 K, respectively. The electrically driven emission of plasmon excitations in 2DEG at 2.7 THz in high electron mobility AlGaN/GaN heterostructures equipped with a grid coupler reaches up to 35 nW when the 2D electron temperature is 225 K in the samples cooled down to 4 K, with WPE of $1\times 10^{-10}$ [22]. The state-of-the-art quantum cascade laser sources equipped with a grating coupler and a metal–metal waveguide provide radiation at 2.8 THz with optical power of 25.4 mW at T=20 K where WPE reaches $1.7\times 10^{-3}$ [39]. RTDs are also a viable option for THz generation, as a 2D array of such devices is able to emit, for example, ∼150 μm and ∼360 nW of power at the frequencies of 0.5 THz and 1.46 THz, respectively [40].

Each of the noted devices has its inherent strengths and weaknesses. For example, QCLs are of high power and electrically driven; however, they possess complicated inner structures, requires low operation temperatures for operation in the THz range, and it is difficult to tune the resonant frequency below a few THz. Plasmon emitters can be made frequency tunable by applying bias voltage between the metal grid coupler and the source contact; however, their output power is quite modest, operation is limited to low temperatures, and advanced processing techniques such as electron beam lithography to manufacture the grid coupler with submicron feature size are required. Moreover, high 2D electron mobility is needed to ensure sharp resonances, leading to increased requirements to structure quality. RTDs operate at room temperature and can easily be assembled to form emitter arrays; however, their inner structure is complex. The AlInAs/GaInAs thermal polaritonic device also contains a complicated inner structure, and it’s lower frequency bound is ∼2.5 THz. On the other hand, the *n*-GaAs/GaAs based and metasurface-equipped thermal emitter discussed in this article is free of the most of limitations noted: (i) it does not require cooling, therefore omitting bulky cryogenic equipment, (ii) the metasurface feature size allows for the use of conventional UV photolitography in its manufacturing, (iii) no complex inner structure is required, as sub-micron deviations in layer thickness are not extremely crucial for emission parameters such as frequency or power, and (iv) it is capable of operating at frequencies below 1 THz, which is important for THz imaging applications.

## 5. Conclusions

In conclusion, GaAs-based magnetic polariton emitters below 2 THz were demonstrated, enabling the integration of such emitters with other GaAs-based THz devices. The RCWA optimized *n*-GaAs/GaAs/TiAu metasurface structures demonstrated predefined resonant lines in the room temperature reflection configuration and in 390 °C temperature emission configuration experiments. The resonance frequency inversely depends on the size of the Ti/Au square elements that make up the metasurface in accordance with the analytical LC model. The first harmonic resonance was measured at 0.7 THz and 1.2 THz for the metasurface with square sizes 42 μm and 26 μm, respectively. The experimentally obtained resonance quality factor values between 3.3 and 5.2 at elevated temperatures are comparable with those of metal-insulator-metal structures and also with quality of 2D plasmon generated resonances at cryogenic temperatures.

## Figures and Tables

**Figure 1 sensors-23-04600-f001:**
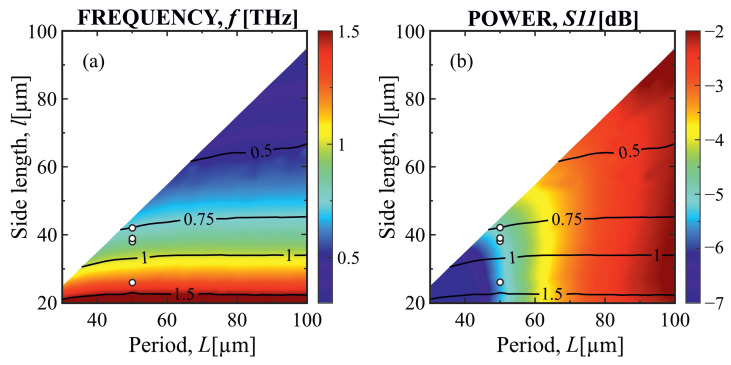
(**a**) Resonant frequency dependency on the period and side length of the metacell. One may note that the resonant frequency is independent of metacell period. This effect exposes the magnetic polariton nature of the observed processes. Emission in the interval from 0.3 THz to 3 THz is available by varying the metacell side length. (**b**) Power at the resonant frequency dependency on the period and side length of the metacell. One may note that the period has significant control over the power of the thermal metacell emitter at the resonant frequency, determining the importance of proper period selection for the creation of an efficient thermal metamaterial-based emitter. The positions of the white dots represent the parameters of the fabricated thermal emitters.

**Figure 2 sensors-23-04600-f002:**
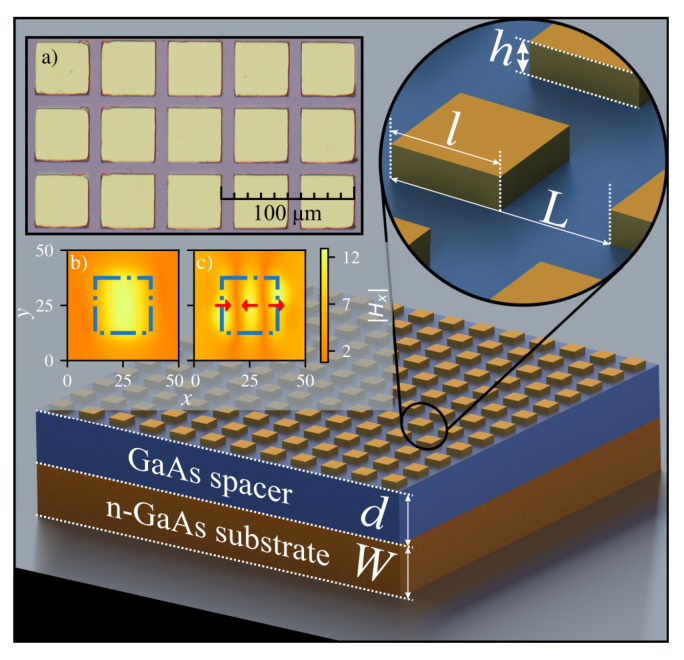
GaAs-based MP excitations sustaining structure that consists of the *n*-GaAs substrate layer (*W*), undoped GaAs spacer layer (*d*), and periodic TiAu metasurface defined by geometrical parameters: the side length (l) of the square, the period (L) of metasurface, and metasurface layer height (h). The inset shows the photo of l=39 μm metasurface acquired with the microscope KH-7700 from *Hirox* (**a**). The absolute value of simulated magnetic field *x* component distributions for the first (**b**) and the third (**c**) MP harmonics under the square metacell. Arrows denote the direction of the flow of the currents inducing the magnetic field. The blue dash-dotted lines represent metalized surface contours.

**Figure 3 sensors-23-04600-f003:**
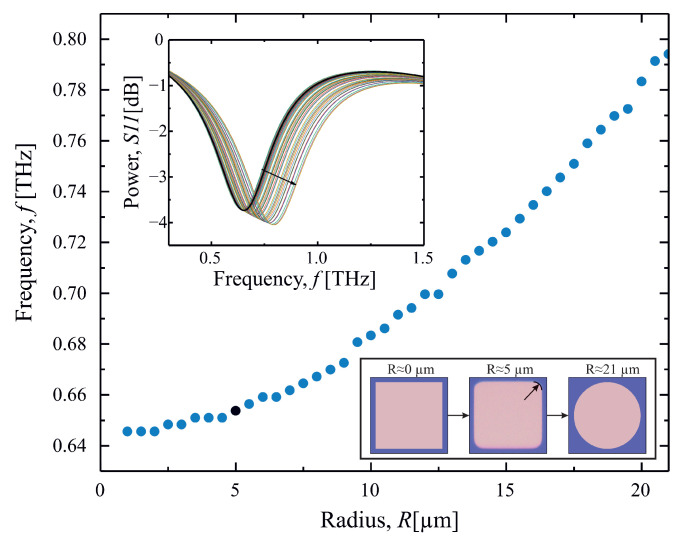
Resonant frequency dependency on the metacell angle fabrication quality, described as the radius of the rounded corners. A small resonant frequency increase is visible with an increase in the radius. Upper left inset depicts spectral dependencies on the rounded corner angle. One may additionally note a small increase in the absorption with a radius increase. Bottom left inset depicts boundary sweep variations being square (R=0μm), circle (R=l/2), and one of the variations (R=5μm), results of which are highlighted by a black point on the main graph and black bold line on the upper left inset.

**Figure 4 sensors-23-04600-f004:**
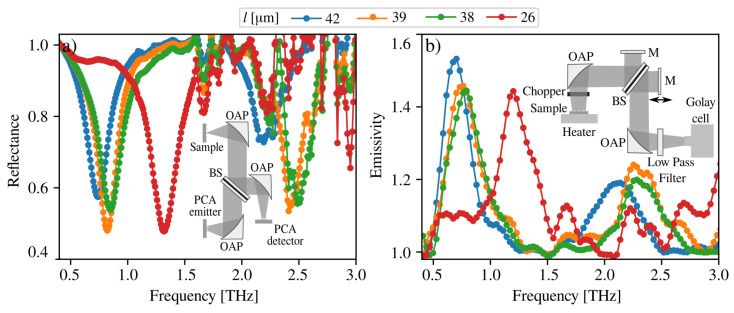
(**a**) Reflectance spectra of GaAs-based metasurfaces with different square metacell side lengths (*l*) at room temperature. Inset: the THz-TDS setup used for reflection measurements. (**b**) GaAs-based metasurface thermo-emission spectra at 390 °C temperature for different square metacell lengths. Inset: the setup of Fourier spectrometer used for emission spectra measurements.

**Figure 5 sensors-23-04600-f005:**
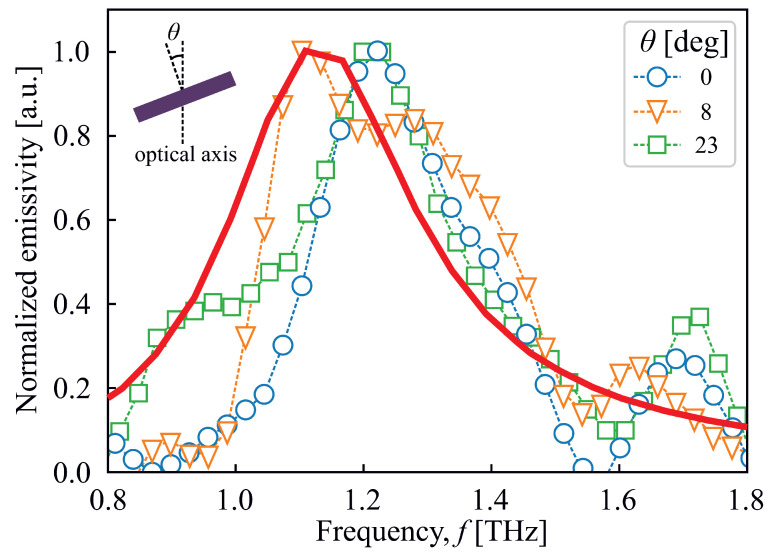
The dependencies of emitted radiation of first MP harmonics on sample rotation angle off the optical axis. Symbols connected with dashed lines denote the different rotation angles: circles—0 deg; squares—8 deg; triangles—23 deg. The solid curve represents the spectrum simulated using the RCWA method. Inset demonstrates the sample orientation during the experiment.

**Figure 6 sensors-23-04600-f006:**
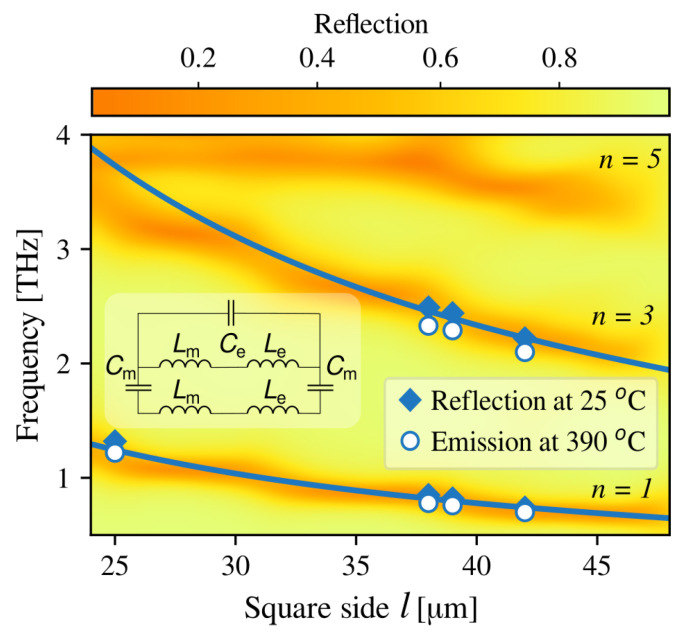
Map of reflection resonant frequency dependence on a rectangular metacell side length calculated by RCWA. Symbols denote the resonant frequencies obtained from experimental reflection (diamonds) and emission (circles) results. The solid curves represent theoretically calculated MPs resonant frequency dependence on single metasurface element side length; *n* denotes the number of the harmonic. The inset shows the LC equivalent circuit used to calculate the theoretical curves.

**Table 1 sensors-23-04600-t001:** Resonant frequency peak values of MPs versus different metacell side lengths estimated in four methods: calculated using the RCWA method, experimentally measured in reflectance configuration, experimentally measured in emission configuration, and calculated using an LC equivalent circuit.

Size, μm	RCWA	Reflectance	Emission	LC
1st harmonic				
42	0.74	0.74	0.71	0.76
39	0.78	0.82	0.77	0.82
38	0.78	0.85	0.79	0.84
26	1.13	1.32	1.22	1.23
3rd harmonic				
42	2.18	2.32	2.11	2.27
39	2.27	2.43	2.32	2.45
38	2.31	2.51	2.32	2.52
26	3.27	–	–	3.69

## Data Availability

The data that support the findings of this study are available from the corresponding author upon reasonable request.

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
