# Peer review of "Narrowband Thermal Terahertz Emission from Homoepitaxial GaAs Structures Coupled with Ti/Au Metasurface"

_sensors, 2023, doi:10.3390/s23104600_

Round 1
Reviewer 1 Report
The paper "Narrowband Thermal Terahertz Emission from Homoepitaxial GaAs Structures Coupled with Ti/Au Metasurface” by I.Grigelionis et al. is devoted to the extremely topical subject of creating integral sources of terahertz radiation.
I think that the work is original and performed at a sufficiently high level and can be published in a journal with minor changes.
1. I would like to recommend add more details about the energy/power characteristics of the proposed source. Including a comparison with existing ones.
2. To analyze the spectral properties of terahertz source, the authors use an equivalent electrical circuit model that explains the characteristic THz radiation peaks. In addition to this model, it would be nice to see some analyze for temperature features. Currently, there are only two measurements at room temperature and at 390 C without deep explanation. But the temperature characteristics must have very important influence for the properties of this source.
Reviewer 3 Report
This paper presents thermal terahertz emission from homoepitaxial n-GaAs/GaAs/TiAu structure. The paper is qualified for publication in Sensors.
This manuscript is well written and of high quality.
1.It is suggest the authors to give some paramters compared with other thz sources in the form of tables or figures in order to show the superiorities.
2.In which kinds of application does the device may be used?
